# A Comparison of Pneumatic and Hand Stripping of Whitefish (*Coregonus lavaretus*) Eggs for Artificial Reproduction

**DOI:** 10.3390/ani10010097

**Published:** 2020-01-08

**Authors:** Radosław Kajetan Kowalski, Beata Irena Cejko, Joanna Grudniewska, Stefan Dobosz, Mirosław Szczepkowski, Beata Sarosiek

**Affiliations:** 1Department of Gamete and Embryo Biology, Institute of Animal Reproduction and Food Research, Polish Academy of Sciences, 10-748 Olsztyn, Poland; b.cejko@pan.olsztyn.pl (B.I.C.); b.sarosiek@pan.olsztyn.pl (B.S.); 2Department of Salmonid Fish Research, Inland Fishery Institute, 83-330 Rutki, Poland; j.grudniewska@infish.com.pl (J.G.); s.dobosz@infish.com.pl (S.D.); 3Department of Sturgeon Fish Breeding, Inland Fisheries Institute, 11-610 Pozezdrze, Poland; m.szczepkowski@infish.com.pl

**Keywords:** whitefish, stripping, pneumatic, reproduction, fertilization, mortality

## Abstract

**Simple Summary:**

This paper describes a technique of pneumatic stripping of whitefish (*Coregonus lavaretus*) eggs with the use of oxygen, nitrogen, and air. This study demonstrates that pneumatic stripping allows the collection of high-quality whitefish eggs. Moreover, we found that gas flow rates should not exceed 0.5 L∙min^−1^ to minimize post-spawning mortality in whitefish. The eggs obtained by pneumatic stripping using oxygen, nitrogen, or air had high hatching rates. We assumed that air stripping is a promising approach for improvement of the whitefish stripping procedure.

**Abstract:**

We describe the technique of pneumatic stripping of whitefish (*Coregonus lavaretus*) eggs with the use of oxygen, nitrogen, and air. Eggs obtained via the traditional method (by pressing the abdominal surfaces) served as a control group. It was established that the gas flow rate during pneumatic stripping should not exceed 0.5 L∙min^−1^, since higher air flow resulted in increased post-spawning mortality. The pneumatic stripping method of egg collection was no faster than hand stripping; however, the time required per female was more consistent. It was found that the pH of the ovarian fluid obtained during hand and pneumatic stripping was not related to the success rate of fertilization. Pneumatic stripping resulted in a higher quality of collected eggs and a higher and more consistent hatching rate as compared with the hand-stripped samples, regardless of the gas used. The results presented here lead us to recommend the pneumatic method for obtaining eggs from whitefish, since it is a simple, reproducible method and improves the reproductive performance and developmental success of the fish eggs.

## 1. Introduction

The whitefish (*Coregonus lavaretus*) belongs to the Salmonidae family, and is one of the most valuable species of freshwater ichthyofauna. It is found in northern and eastern Europe, for example, Great Britain, Finland, and Russia. In Poland, the whitefish appears in two forms, lake dwelling (in oligotrophic lakes) and migratory fish. It has also been confirmed that only two native populations of whitefish reproduce naturally in Polish waters, namely Pomorska Bay and Lake Łebsko [1]. The reproductive cycle of the whitefish begins in autumn, and lasts from the second half of October until the end of December, during which time the water temperature varies between 2 and 6 °C with an average of 4.5 °C. The whitefish is very sensitive to water quality [2,3,4] and due to water pollution, overfishing, and a limited number of spawning sites, the natural reproduction rate of whitefish has decreased significantly in recent years [5]. For example, in Poland, the native population of whitefish from Lake Łebsko has become endangered, and therefore a conservation project supervised by the Słowiński National Park Authority was conducted for several years [6]. In order to increase the whitefish population, sperm analyses have been carried out and improvements have been made in its quality control [7,8], sperm storage and cryopreservation [9,10], and the production of stocking fry under controlled conditions [11,12]. However, work focused on increasing the egg developmental success in whitefish has not yet been done. These activities are essential, because the conservation of aquaculture is a very important tool for the restitution and support of fish species that are in danger of extinction. They are also important due to changes in the climate that have been observed in recent years, and their effects on reducing the occurrence of wild species like whitefish [12]. Despite the fact that the wild population of whitefish is decreasing, the production and sale of whitefish have increased in Europe from 2344 t in 2000 to 5310 t in 2017 [13] and this species has recently attracted considerable interest from fish producers [14]. This fish undoubtedly represents high value to consumers, and therefore production of this species could offer a valuable addition to commercial trout-breeding facilities [15].

Under hatchery conditions, whitefish eggs are collected manually, i.e., using hand stripping, which involves manually massaging the abdomen of the female to strip off the ova. This method is effective, although it has been confirmed that in salmonid fish, for example, rainbow trout (*Oncorhynchus mykiss*), this procedure can result in broken eggs, which decreases the pH of the ovarian fluid and releases egg yolk, both of which negatively affect egg fertilization [16]. Hence, to improve the quality of the obtained gametes in hatchery practice, alternative methods of egg collection such as pneumatic stripping can be used [17]. This method is based on the injection of gases (air, nitrogen, or oxygen) into the body cavity to expel eggs using gas pressure. Pneumatic methods of egg collection were first used in the 1960s. They have been successfully applied with no negative side effects in salmonid species and are quicker and easier for both the females and the stripper [18]. It has been confirmed that in northern pike (*Esox lucius*) the application of a pneumatic method resulted in higher percentages of fertilized eggs and hatched larvae as compared with the traditional method of egg collection (hand stripping) [19]. In recent studies, air and hand stripping methods have also been studied with respect to the one-year survival rate and egg quantity and quality in two types of salmonid fish, rainbow trout and brown trout (*Salmo trutta morpha fario*) [20]. The results indicated that air stripping yields better quality eggs and higher one-year survival rates in rainbow trout. In addition, air stripping resulted in lower mortality rates than hand stripping (25% vs. 35%).

The aim of this study is to test whether the pneumatic method of egg collection is a beneficial tool for the reproduction of whitefish under controlled conditions. Herein, in contrast to the earlier work [20], to determine the optimal conditions during egg stripping, various flow rates (0.5, 1.0, and 1.5 L per min) and several types of gas (oxygen, nitrogen, and air) were investigated. The gases sources were chosen based on their properties and availability in common practice. Nitrogen is expected to create conditions in which the oxidation process is reduced, moreover, it is a relatively cheap source of gas. Oxygen is a common gas used in hatcheries; however, might have a negative effect on the eggs due to increased oxidation and generation of reactive oxygen species. Air was chosen due to its limited oxidation potential and as the cheapest source of gas. The effectiveness of pneumatic stripping was based on the fertilization rate at the eyed stage, egg yield, and hatching rate after its application to whitefish under controlled conditions.

Procedures were carried out in accordance with the Local Committee on the Ethics of Animal Experiments in Olsztyn, Poland no 24/2011/N.

## 2. Materials and Methods

### 2.1. Broodstock Origin and Characteristics

A broodstock of whitefish at two years of age, with an average weight of 632.7 ± 61.4 g, were obtained from the Department of Salmonid Research and Department of Sturgeon Breeding (Inland Fishery Institute in Olsztyn, Poland). Before stripping the females, maturity was checked by visual inspection of the eggs, which were freely expelled by the fish during the inspection. Only the fish that were evaluated as being mature were used in the experiment. To ensure identification, all fish used in the experiment were marked with passive integrated transponder (PIT) tags. In total, 112 females and 4 males were used in experiments.

### 2.2. Experimental Design

The first experiment analyzed the effects of gas pressure on the broodstock mortality. The 80 fish used were divided into four groups and subjected to the air stripping procedure with air as a gas source at three different rates of gas flow. A control group was hand stripped. The rate of mortality was analyzed in this experiment.

The second experiment analyzed the effects of the type of gas (air, oxygen or nitrogen) using the flow that gave the lowest mortality (0.5 L∙min^−1^). Thirty-two fish were divided into four groups (hand stripped, air, oxygen, and nitrogen stripped). During this experiment, we recorded the time of stripping, the relative fecundity of the fish, the pH of the ovarian fluid, and the fertility of the obtained eggs.

### 2.3. Stripping Procedure

Before egg collection, the fish were anaesthetized with Propiscin (Inland Fishery Institute, Olsztyn, Poland) at a dosage of 0.7 mL∙L^−3^ [21]. The hand and pneumatic stripping procedures were carried out by the same operator. In order to obtain eggs using gas stripping, a 0.8 mm needle was inserted under the pectoral fins. This needle was attached to a gas-dosing device using a thread to prevent the needle from falling out. The eggs were collected separately from each female and placed in a dry plastic bowl. The hand-stripping process for the whitefish eggs is shown in Figure 1A,B, and the pneumatic stripping is illustrated in Figure 1C,D.

### 2.4. Effect of Different Gas Flows on the Broodstock Mortality

To determine the effect of air flow on the broodstock mortality, air stripping was performed with equipment that allowed control of the gas source, pressure, and flow rate. Air was used as a gas source, and the gas pressure was 0.8 bar with a flow rate of 0.5, 1.0, or 1.5 L∙min^−1^. Each flow rate was tested by stripping eggs from 20 fish (in each group), and the mortality was recorded every hour for 24 h after spawning.

### 2.5. Effects of Different Gas Sources on Pneumatic Stripping

The fish were divided into the following four groups: In three groups, pneumatic stripping was applied using different types of gas, i.e., oxygen (*n* = 8), nitrogen (*n* = 8), and air (*n* = 8); in one group of females, hand stripping was applied using gentle abdominal massage of the fish (*n* = 8). The average weights of the fish in each group were not statistically different. The following parameters were collected for each group:

The total time (s) of stripping, measured from the release of the first eggs to the release of the fish from the spawning ledge;

The relative fecundity (RF) of the fish = 100  × We/Wf. *We* = weight of eggs, *Wf* = weight of the female.

The pH of the ovarian fluid, analyzed using an Orion Ross Ultra Electrode (Thermo Scientific, Waltham, MA, USA). The fertility of the obtained eggs, measured by the percentage of the eyed eggs and hatching rate of 100 eggs incubated following collection using formulas:(1)$$\mathrm{Fertility}\;\mathrm{at}\;\mathrm{eyed}\;\mathrm{stage}\;=\;\frac{\mathrm{no}.\;\mathrm{of}\;\mathrm{eyed}\;\mathrm{eggs}}{\mathrm{total}\;\mathrm{number}\;\mathrm{of}\;\mathrm{eggs}}\times 100,$$
and
(2)$$\mathrm{hatching}\;\mathrm{rate}\;=\;\frac{\mathrm{no}.\;\mathrm{of}\;\mathrm{hatched}\;\mathrm{eggs}}{\mathrm{total}\;\mathrm{number}\;\mathrm{of}\;\mathrm{eggs}}\times 100.$$

The fertility measurements were carried out as duplicates.

### 2.6. Fertilization Procedure

Sperm with high motility (above 80%), obtained from four males, was pooled and used for fertilization. The concentration of sperm was measured using a Bürker chamber. The eggs were fertilized with about 100,000 sperm per egg, and about 100 eggs per female were fertilized. After the introduction of sperm, the eggs were gently stirred with a feather. Hatchery water was added to the eggs and the sperm and was poured out after about 2 min, followed by rinsing with an aqueous solution of tannin to reduce the viscosity. Next, the solution was removed, and the eggs were rinsed with water and placed in a grid (each experimental variant was placed separately). Eggs incubator supplied with river water of pH ranged from 7.2 to 7.5 and temperature of 4 ± 2 °C during the incubation period. The numbers of eyed eggs and larvae resulting from the total numbers of eggs and larvae were counted in each experiment. The results are presented as a mean ± SEM.

### 2.7. Statistical Analysis

All analyses were performed at a significance level of 0.05 using GraphPad Prism 8.0 (GraphPad Software Inc., San Diego, CA, USA). For the comparison of survival curves, we used a log-rank (Mantel–Cox) test. For statistical procedures, the data percentages were normalized using an arcsine square root transformation. The data were analyzed using a repeated measure one-way analysis of variance (ANOVA), followed by Tukey’s post-hoc test.

## 3. Results

### 3.1. Effect of Gas Flow on the Broodstock Mortality

The fish mortality was higher after air stripping with an air flow rate of 1.0 and 1.5 L∙m^−1^ than that observed after hand stripping (Figure 2, *p* < 0.001). The mortality rate following air stripping with a flow rate of 0.5 L∙m^−1^ showed no significant difference as compared with that of the hand-stripped group.

### 3.2. Effect of Gas Source on Stripping Time, Relative Fecundity, and pH of Ovarian Fluid

The duration of stripping was similar in all groups and did not exceed 1 min regardless of the method used. The mean values of the stripping times ranged between 17 s (pneumatic stripping using nitrogen) and 25 s (hand stripping) (Figure 3A, *p* > 0.05). The relative fecundity indicated by the weight of the eggs did not differ significantly between the groups. The mean values ranged from 15.6% (pneumatic stripping using air) to 20.4% (hand stripping) (Figure 3B, *p* > 0.05). The method of egg collection also did not influence the pH of the ovarian fluid of whitefish females. The mean values of this parameter were between 8.3 (pneumatic stripping using air) and 8.1 (hand stripping) (Figure 3C, *p* > 0.05). The mortality measured 24 h after spawning was as follows: hand stripped group 25%, air group 37.5%, oxygen group 25%, and nitrogen group 12.5%. There were no statistically differences in mortality rate among the groups.

### 3.3. Effect of Gas Source on Eyed Eggs and Hatching Rate

The lowest percentage of eyed eggs (32.4%) and the lowest hatching rate (28.0%) were observed for groups in which eggs were collected using hand stripping (Figure 4A,B). When pneumatic stripping was used for egg collection with oxygen, nitrogen, and air, the eyed eggs and the hatching rates reached values of 53.9% and 42.3%, 55.5% and 44.5%, and 53.7% and 45.8%, respectively (Figure 4B, *p* < 0.05). The highest variation, of both fertilization at the eyed stage and hatching rate, were observed in the hand stripped group.

## 4. Discussion

The results of this study demonstrate, for the first time, that under hatchery conditions, eggs from whitefish can be collected using pneumatic stripping, and different gases (i.e., oxygen, nitrogen, and air) with a flow rate of approximately 0.5 L·min^−1^ can be successfully utilized for this procedure. No significant differences were observed between hand and pneumatic stripping in terms of the time of stripping, the relative fecundity of the females, and the pH of the ovarian fluid. However, based on the percentage of eyed eggs and the hatching rate, the lowest reproductive performance was observed following hand stripping. This indicates that pneumatic stripping could be a more appropriate and efficient method of egg collection than the abdominal massage of whitefish under artificial conditions.

One of the problems with whitefish reproduction under controlled conditions is post stripping mortality, because females are very sensitive to hatchery manipulations [22]. Therefore, to determine the most appropriate gas flow and to prevent excessive female mortality, different air flow rates of 0.5, 1.0, and 1.5 L·min^−1^ were applied in egg collection. A high mortality rate was observed when the highest air flow (1.5 L∙min^−1^) was used for stripping of the whitefish, but this mortality rate decreased significantly when a lower air flow was applied (0.5 L∙min^−1^) and reached a similar mortality rate to that of hand stripping. The air flow applied in this study was similar to the air flow used in our earlier study of egg collection in northern pike using the stripping method [19]. However, in rainbow trout and brown trout, the most efficient gas flow used was 1.5 L·min^−1^ [20]. The differences between the gas flow rates used for these species depend on the physiology of reproduction (i.e., ovulation into the body cavity or ovaries) and the size of the eggs (large or small). A high flow rate can result in post-stripping mortality of the females, while a slower speed can unnecessarily prolong the procedure and result in not obtaining all the eggs; thus, the gas flow rate should be determined before stripping for each species studied.

The time spent on oocyte collection varied between approximately 17.0 and 19.3 s for pneumatic stripping and was almost 25 s for hand stripping. It is noteworthy that although the highest variation was observed for samples obtained using hand stripping, the collection of salmonid eggs typically takes longer using pneumatic stripping than hand stripping [20]. In the case of whitefish, we did not observe a prolonged time for pneumatic stripping for any of the gases tested (i.e., oxygen, nitrogen, and air). Despite the similar times required for both procedures, the time spent on each fish varied less when pneumatic stripping was used. This is an important result, as it allows producers to estimate the necessary spawning time before reproduction, which could facilitate better organization of hatcheries. Moreover, the gas source has no effects on the relative fecundity, pH of the ovarian fluid, eyed eggs, and hatching rate. This might be due to the very short exposition of the gases during the procedure. Therefore, any expected benefits of nitrogen use (lack of reactive oxygen species generation) or negative effects of oxygen (higher oxidation rate) are not seen in the results of the stripping. The relative fecundity was similar for all groups, and no evidence of increased effectiveness was observed for the air stripping procedure. The values recorded in the present study were similar to those reported by Szczepkowski et al. [22] and were slightly higher than those by Kankainen et al. [23], who indicated an average egg yield of around 16% from whitefish. Despite these differences, the whitefish has high potential in terms of egg production. Since breeding programs are typically related to the quantity and quality of flesh [24], programs aimed at increasing the caviar production efficiency of whitefish are being considered in Finland [23]. The pH of the ovarian fluid is an indicator of egg quality in salmonid fish, with a higher pH level relating to higher quality eggs [16,20,25]. Although this is true for trout eggs, we did not observe a similar correlation in our study. In rainbow trout, the lowest pH levels for ovarian fluid were reported for a group in which spawning was conducted by hand [20]. In this study, hand stripping was carried out gently, so that the quality of eggs did not differ statistically from the quality of eggs obtained via the pneumatic stripping. Similar results have been reported for northern pike [19]. It is important to note that the hand stripping method is completely dependent on the operator experience and could result in severe damage to the eggs if conducted without necessary care.

The fertilization rate at the eyed stage and hatching rate was higher in the pneumatic stripping group, regardless of the gas used for egg collection, and we found that the hand-stripped group exhibited the highest variability in fertilization success. These results suggest that hand stripping method could negatively affect the reproductive performance of whitefish, as it was shown for the rainbow trout earlier [20]. The high variability of the hand stripped eggs quality could be connected with the individual characteristics of whitefish females. Some female could produce eggs different in size [26] and possibly softer eggs, and thus become more vulnerable to damage caused by human hands during stripping. Moreover, the differences in the volume of ovarian fluid which could protect the eggs from excessive pressure during hand stripping also might negatively affect the hand stripping outcomes. Fewer fluids usually make hand stripping procedure more difficult and involve more power from the operator. Thus, the possibility of damaging eggs increases, which subsequently could result in a lower hatching rate. All these taken together, suggest that the hand stripping method, even when carefully performed, could result in high variability of the obtained eggs quality.

Unlike in rainbow trout [20], there was no correlation between the pH of ovarian fluid and survival rate at the eyed stage nor at hatching. This observation corroborates the results of Svinger and Kouril [26], who did not note any correlation between the pH of ovarian fluid and survival rate at the eyed stage and hatching in northern whitefish (*Coregonus peled*). The lower hatching rate observed in hand-stripped samples indicated that whitefish eggs are susceptible to mechanical damage during the spawning procedure, as has been shown for northern pike [19] and rainbow trout [20]. Our hatching rates did not exceed 50%, which is in line with reports from other research groups [26,27]. This low efficiency of reproduction may be associated with the high susceptibility of whitefish to changes in temperature [28]. Indeed, during incubation in river water we noted relatively high-temperature fluctuation (from 2 to 6 °C). These thermally unstable conditions could be responsible for the low hatching rate achieved in our study. Taken together, the data obtained in our study and those of other researchers indicate that the reproductive biology of whitefish needs to be further studied for better management of hatchery broodstock.

The use of air stripping in different species needs specific adjustments as the anatomical and physiological differences between fish affects the procedure parameters. Whitefish as compared with rainbow trout or brown trout have clear differences which reflect in their higher susceptibility to the stress as well as lower ovarian fluid volume, higher relative eggs volume, and smaller and softer eggs. These differences lead to specific changes in air-stripping procedure such as lower gas flow and smaller needle diameter. These results offer some indications for further air stripping method adjustments in term of its expansion on the other species.

In summary, the results of pneumatic stripping reveal the conditions that are required to obtain high quality eggs, as characterized by higher fertilization success than those obtained by manual stripping. It should be stressed that pneumatic stripping, which preserves the natural mucus of fish, may be beneficial in terms of successful long-term broodstock management. Another advantage of this method is the potential for standardization, as the only influential factors are the air flow rate and amount of eggs in the body of the fish. The efficiency of pneumatic stripping is related only to air pressure and not to the skill of the operator, and therefore is highly recommended in hatcheries, where the availability of skilled workers is limited. Increased data from future studies could support the use of air stripping for other fish species.

## 5. Conclusions

The air stripping methods of fish eggs collection could be adopted in other than trout species. In whitefish, this method allows for obtaining higher quality eggs as compared to hand stripped eggs. Contrasting to trouts [20] where 1.5 L·min^−1^ gas flow was optimal, in whitefish we have limited this parameter to 0.5 L·min^−1^ due to high mortality when the faster gas flow was used. Moreover, after applying different gas sources such as air, nitrogen and oxygen, during the air stripping procedure, we did not find differences in any measured parameters. Therefore we assume the type of gas is not important in the application of the air stripping method in whitefish reproduction. We conclude that the air stripping method of egg collection can improve the efficiency of the whitefish stripping procedure.

## Figures and Tables

**Figure 1 animals-10-00097-f001:**
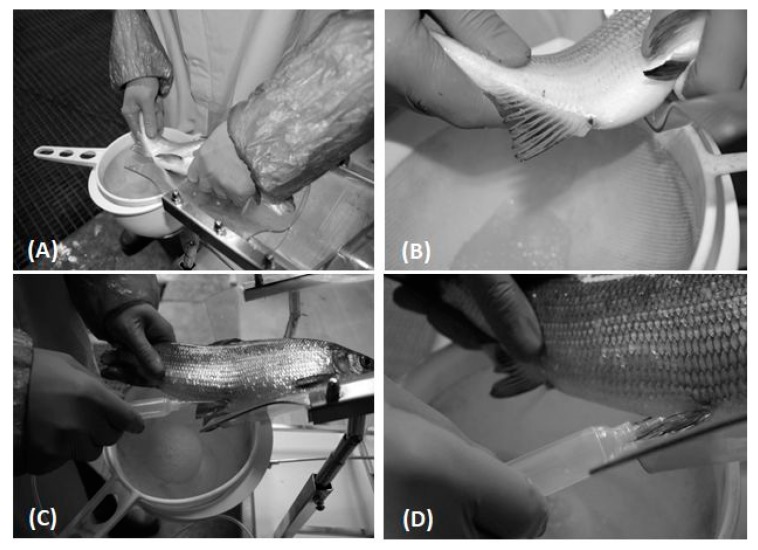
Comparison of hand (**A**,**B**) and pneumatic (**C**,**D**) stripping method of whitefish (*Coregonus lavaretus*) egg collection under controlled condition.

**Figure 2 animals-10-00097-f002:**
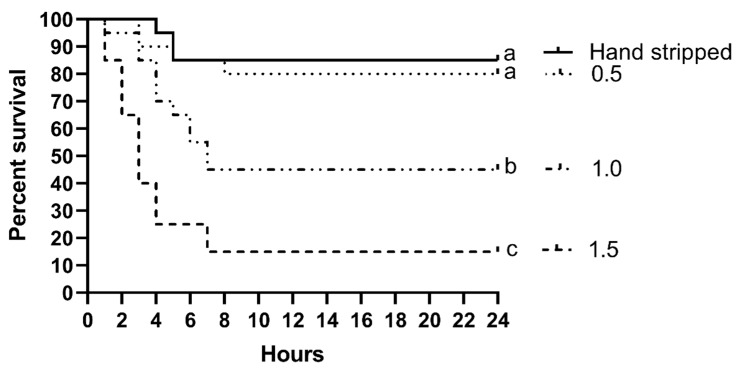
Fish survival proportions represented as Kaplan–Meyer curves. Fish were divided into 4 groups: Hand stripped, stripped with air flow of 1.5, 1, and 0.5 L·min^−1^. The *p*-value for log-rank (Mantel–Cox) test < 0.0001 and *p*-value for log-rank test for trend = 0.7038.

**Figure 3 animals-10-00097-f003:**
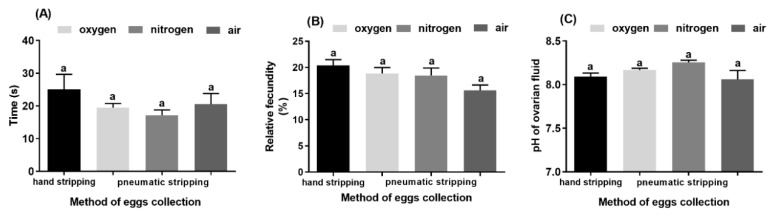
Stripping time, relative fecundity, and ovarian fluid measured following hand and pneumatic stripping with oxygen, nitrogen, and air. (**A**) Time required for egg collection, (**B**) relative fecundity of fish measured after stripping, and (**C**) pH values of the ovarian fluid obtained during spawning (*p* < 0.05). Data presented as mean and standard error of mean (±SEM). Values marked with the same letter indexes showed no significant differences (*p* > 0.05).

**Figure 4 animals-10-00097-f004:**
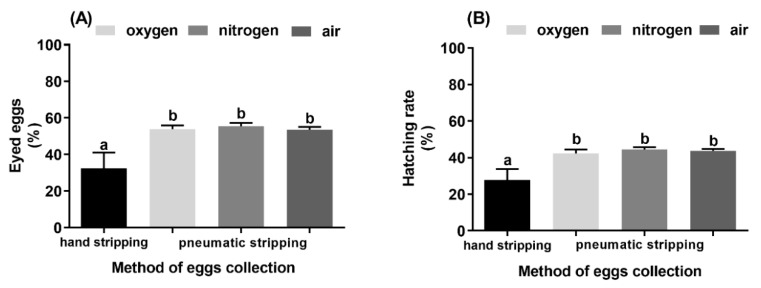
Fertilization success of the eggs obtained by hand stripping and using the pneumatic method with oxygen, nitrogen, and air. (**A**) The percentage of the eyed eggs and (**B**) the hatching rate. Data presented as mean and standard error of mean (± SEM). Values marked with different letter indexes showed significant differences (*p* < 0.05).

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
