# Peer review of "A Comparison of Pneumatic and Hand Stripping of Whitefish (*Coregonus lavaretus*) Eggs for Artificial Reproduction"

_animals, 2020, doi:10.3390/ani10010097_

Round 1

Reviewer 1 Report

The authors described a pneumatic method of different gases and different pressures to retrieve Whitefish eggs.

L116: Please write a subtitle of 2.4. before subheadings 2.4.1 & 2.4.2.

L115-118: remove italic format.

L185-186: incorrect, significant differences (p < 0.05).

Author Response

Rewiever 1

Comments and Suggestions for Authors

The authors described a pneumatic method of different gases and different pressures to retrieve Whitefish eggs.

L116: Please write a subtitle of 2.4. before subheadings 2.4.1 & 2.4.2.

Corrected, since our subheadings were wrongly applied, we have changed numbers in this part of the manuscript.

L115-118: remove italic format.

Corrected, the italic format has been removed from the text.

L185-186: incorrect, significant differences (p < 0.05).

Corrected, the signature under Fig. 4 has been changed.

Reviewer 2 Report

This paper describes the results of different methods used to obtain eggs from females of whitefish. The length of the paper is adequate although formulas could be rewritten. The data presented are relevant and of great interest for the artificial reproduction of several species. It is essential to improve the techniques that increase the efficiency of the methods and the health of the fish.

Please consider revision of the manuscript according to the following comments:

Scientific name of whitefish in cursive in title and summary

Superscript in line 18, 24

Scientific names in cursive line 36, 61, 69, 73

Delete comma line 71.

Line 73-75. Are results of this work?, or of the work previously cited? Then, “The results indicated….”

Line 76…….  Why are these gases chosen? Is there any reason?

Could the authors provide some explanation or information about it?

Line 77-79: Could it be better to reverse the sentence? Gas flow first and type of gas later? So it would be according to the subsequent order.

Line 115-118: correct font style and tabulation.

From line 123 to 131: the authors could consider the following changes

Parameters instead of data (The following data were..) The total time (s) instead of “the total time of stripping, measured from..” The relative fecundity (RF) of the fish = 100 x We/Wf, We = weight of eggs, Wf = weight of the female; instead of “the relative fecundity of the fish, measured as: ?? = ?? ?? × 100, where RF = relative fecundity, We = weight of eggs, Wf = weight of the female;” The Ph instead of “the pH” Move from 129 to line 130: “the fertility of the obtained eggs, ….” And change: The fertility…. In line 130: … “rate of 100 eggs incubated following collection using the formulas:” and delete the point. Delete (1) (2) (3), It is not known what it means. Could the authors change the format of the formulas in the same way as the previous one?

Move the line 132 to following paragraph?

In general: check decimal notation.

Line 152: change to lower case (P < 0.001)

Line 167: delete “important”

Figure 4 caption: “Values marked with different letter indexes showed no significant differences (p > 0.05)”. It is right? or it is to the contrary?

Line 234: delete 1 tabulation

From line 320 to line 345 delete double numbering of references and font size of reference 9.

Author Response

Reviewer 2

Comments and Suggestions for Authors

This paper describes the results of different methods used to obtain eggs from females of whitefish. The length of the paper is adequate although formulas could be rewritten. The data presented are relevant and of great interest for the artificial reproduction of several species. It is essential to improve the techniques that increase the efficiency of the methods and the health of the fish.

Please consider revision of the manuscript according to the following comments:

Scientific name of whitefish in cursive in title and summary

Corrected, the Latin name of the whitefish was provided in italic throughout the text.

Superscript in line 18, 24

Corrected, the record has been changed.

Scientific names in cursive line 36, 61, 69, 73

Corrected, scientific names were provided in italic throughout the text.

Delete comma line 71.

Corrected.

Line 73-75. Are results of this work?, or of the work previously cited? Then, “The results indicated….”

Corrected.

Line 76…….  Why are these gases chosen? Is there any reason? 

Could the authors provide some explanation or information about it?

We have added the explanation.

Line 77-79: Could it be better to reverse the sentence? Gas flow first and type of gas later? So it would be according to the subsequent order.

Corrected.

Line 115-118: correct font style and tabulation.

Corrected.

From line 123 to 131: the authors could consider the following changes

Parameters instead of data (The following data were..) The total time (s) instead of “the total time of stripping, measured from..” The relative fecundity (RF) of the fish = 100 x We/Wf, We = weight of eggs, Wf = weight of the female; instead of “the relative fecundity of the fish, measured as: ?? = ?? ?? × 100, where RF = relative fecundity, We = weight of eggs, Wf = weight of the female;” The Ph instead of “the pH” Move from 129 to line 130: “the fertility of the obtained eggs, ….” And change: The fertility…. In line 130: … “rate of 100 eggs incubated following collection using the formulas:” and delete the point. Delete (1) (2) (3), It is not known what it means. Could the authors change the format of the formulas in the same way as the previous one?

Corrected.

Move the line 132 to following paragraph?

We are unsure what this comment stand for. Line 132 was the title of subheading, however, due to other reviewer comments, we have changed it into headings.

In general: check decimal notation.

Indeed we have found two errors in decimal notation. Corrected.

Line 152: change to lower case (P < 0.001)

Corrected.

Line 167: delete “important”

Corrected.

Figure 4 caption: “Values marked with different letter indexes showed no significant differences (p > 0.05)”. It is right? or it is to the contrary?

Corrected, we change the signature of the Fig. 4.

Line 234: delete 1 tabulation

Corrected.

From line 320 to line 345 delete double numbering of references and font size of reference 9.

Corrected, the doubled numbering of references was deleted and the font size of reference 9 was used.

Reviewer 3 Report

Major Comments

Please clearly show the difference with the Previous JoVE paper  (Ref.20).

I  only know the difference with the fish species.

Minor Comments

L55-L58: No connection with previous sentences.

L112-115: Why do you write in Italic?

L146-L168: There are no Fig.

L302-L328. Refer.no. are overlap.

Author Response

Reviewer 3

Comments and Suggestions for Authors

Major Comments

Please clearly show the difference with the Previous JoVE paper  (Ref.20).

I  only know the difference with the fish species.

We have added the explanation.

Minor Comments

L55-L58: No connection with previous sentences.

We rewrote the sentence for better clarity.

L112-115: Why do you write in Italic?

That was our mistake, we corrected this part of the manuscript.

L146-L168: There are no Fig.

We are unable to address this question. It was probably editing mistake.

L302-L328. Refer.no. are overlap.

That was our mistake, we corrected this part of the manuscript.

Round 2

Reviewer 3 Report

In the first place, this manuscript is carbon copy (only change species) of their previous reports (IF is lower than [Animals]).

Thus, I recommend you, unfortunately, to reject this manuscript.